# Synergistic Activity of Tetrandrine and Colistin against *mcr-1*-Harboring *Escherichia coli*

**DOI:** 10.3390/antibiotics11101346

**Published:** 2022-10-02

**Authors:** Muhammad Shafiq, Fen Yao, Hazrat Bilal, Sadeeq Ur Rahman, Mi Zeng, Ilyas Ali, Yuebin Zeng, Xin Li, Yumeng Yuan, Xiaoyang Jiao

**Affiliations:** 1Department of Cell Biology and Genetics, Shantou University Medical College, Shantou 515041, China; 2Department of Pharmacology, Shantou University Medical College, Shantou 515041, China; 3Guangdong Provincial Key Laboratory of Infectious Diseases and Molecular Immunopathology, Shantou University Medical College, Shantou 515041, China; 4Department of Dermatology, The Second Affiliated Hospital of Shantou University Medical College, Shantou 515041, China; 5Department of Microbiology, Abdul Wali Khan University, Mardan 23200, KP, Pakistan; 6Department of Medical Cell Biology and Genetics, Health Science Center, Shenzhen University, Shenzhen 518060, China

**Keywords:** *E. coli*, colistin resistance, *mcr-1*, tetrandrine, synergistic activity, docking

## Abstract

Before the emergence of plasmid-mediated colistin resistance, colistin was once considered the last drug of choice for infections caused by carbapenem-resistant bacteria. Currently, researchers are relentlessly exploring possible alternative therapies that could efficiently curb the spread of drug resistance. In this study, we aim to investigate the synergistic antibacterial activity of tetrandrine in combination with colistin against *mcr-1*-harboring *Escherichia coli*. We examined the antibacterial activity of tetrandrine in combination with colistin in vivo and in vitro and examined the bacterial cells by fluorescence, scanning, and transmission electron microscopy (TEM) to explore their underlying mechanism of action. We further performed a computational analysis of MCR-1 protein and tetrandrine to determine the interaction interface of these two molecules. We confirmed that neither colistin nor tetrandrine could, on their own, inhibit the growth of *mcr-1*-positive *E. coli*. However, in combination, tetrandrine synergistically enhanced colistin activity to inhibit the growth of *E. coli* both in vivo and in vitro. Similarly, molecular docking showed that tetrandrine interacted with the three crucial amino acids of the MCR-1 protein in the active site, which might inhibit MCR-1 from binding to its substrates, cause MCR-1 to lose its ability to confer resistance. This study confirmed that tetrandrine and colistin have the ability to synergistically overcome the issue of colistin resistance in *mcr-1*-harboring *E. coli*.

## 1. Introduction

*Escherichia coli* is a gram-negative bacterium that causes infections in humans, poultry, and animals. In typical cases, antibiotics can kill or inhibit the growth of *E. coli*. However, after acquiring antibiotic-resistance genes, their treatment with routinely used antibiotics becomes difficult. Different resistance mechanisms have been investigated in *E. coli* [1,2,3,4], among which extended-spectrum β-lactamase (ESBL) production due to various genes, such as *bla*_CTX-M_, *bla*_TEM_*,* and *bla*_SHV_, has been most studied [5,6]. In such cases, the carbapenems were considered the best therapeutic options prior to the development of resistance, due to improper use, by acquiring *bla*_NDM_, *bla*_KPC_, *bla*_OXA_, and many other resistance genes [7,8,9]. As a result, colistin became one of the last-resort antibiotics against these superbugs. Initially, only chromosomal-based resistance was developed against colistin. Nonetheless, it was not much of a threat, because it was only transmitted vertically [10]. However, in late 2015, the first plasmid-mediated colistin resistance gene, mobilized colistin-resistance (*mcr-1*), was detected in China in pigs and humans; this initiated the post-antibiotic era, because all available antibiotic options were countered by bacterial resistance mechanisms [11]. Nowadays, carbapenem-resistant *Enterobacterales* harboring *mcr* have been reported worldwide [12,13,14]; therefore, colistin is alternatively used to cure those infections that lead to increases of the colistin resistance [15,16].

Researchers from various regions around the world have been investigating many alternative ways to cure such superbugs, and bacteriophage therapy, nano-medicines, and herb extracts have shown good activity. Colistin was also considered to be one of the last-resort treatments used to treat infections caused by multidrug-resistant bacteria [17]. Due to the narrow therapeutic action and nephrotoxicity of polymyxin, several recent studies have used a combination of colistin with other existing antibiotics and herbal medicines to find alternative therapies to eradicate colistin resistance and slow down its dissemination [18,19,20,21]. Some recent studies found that drug-combination therapies could be successful by using drug-repurposing strategies [22,23]. Among these, approaches using extracts of natural herbs are easy and have fewer side effects [24]. The alkaloid extract of *Stephania tetrandra* plant named “tetrandrine”, with the chemical structure “6,6,7,12-tetra methoxy-2,2” [25], is one of the herbal extracts that have many beneficial aspects against tumors, *Candida* mycosis, heart arrhythmia, and blood pressure.

The synergistic antibacterial activity of tetrandrine with other drugs has already been reported against *Staphylococcus aureus, Mycobacterium smegmatis*, and *Salmonella* species [26]. It has been reported that tetrandrine has an inhibitory effect on the efflux pump of bacteria, which helps in the absorption and retention of drugs inside the bacterial cells and increases their antibacterial activity [27]. In the present study, we investigate its effect in combination with colistin against the plasmid-mediated *mcr-1*-harboring, colistin-resistant *E. coli* isolated from the feces of healthy livestock in Pakistan. Their synergism was determined by checkerboard assay, and further analyses of its activity were carried out through electron microscopy. The findings of this study could provide a baseline for further research work in order to establish an alternative therapeutic option for treating colistin-resistant *E. coli* infections.

## 2. Results

### 2.1. Antibacterial Activity

The antibacterial activity of colistin and tetrandrine, alone and in combination, was determined. All *mcr-1*-positive isolates had a MIC ≥ 2 µg/mL and showed resistance to colistin. Similarly, the MIC of tetrandrine was found in the range of 320 to 1280 µg/mL, indicating that tetrandrine alone is not a suitable inhibitor of the growth of *mcr-1*-positive isolates. Interestingly, when colistin and tetrandrine were combined in the checkerboard assay, the FICI values were recorded as <0.5 for all 20 tested *mcr-1*-positive isolates, indicating a synergistic activity between tetrandrine and colistin. Similarly,the MIC concentrations of tetrandrine and colistin were applied against the tested isolates. We found a ≥ 4- to ≥ 256-fold increase in the MIC for colistin (Table 1). Similarly, the time-kill assay showed that neither colistin nor tetrandrine alone reduced the growth of *E. coli*, but, with combination treatment, a dramatic reduction in bacterial growth was recorded (Figure 1). Similar results were observed with fluorescence microscopy (Figure 2). These results suggest that tetrandrine enhanced the activity of colistin and inhibited the growth of *mcr-1*-harboring, colistin-resistant *E. coli*.

### 2.2. Scanning Electron Microscopic Observation

The SEM observation was performed to examine the morphological and membrane-structure modification of *mcr-1*-harboring *E. coli* to explore the synergistic antibacterial activity of colistin and tetrandrine. Bacteria were incubated for 4 h with colistin (2 µg/mL) and tetrandrine (MIC value), alone and in combination. Micrographs show untreated *E. coli* (Figure 3A), *E. coli* treated with colistin alone (Figure 3B), and *E. coli* treated with tetrandrine alone (Figure 3C). The untreated cells and cells treated with colistin or tetrandrine alone had an unvarying rod shape with an intact membrane, smooth surface, and no visible morphological modifications. On the other hand, multiple dents, holes, and deep hollows were seen in the colistin-plus-tetrandrine-treated bacteria (Figure 3D). Many protruding and distorted cells, and some permanently lysed cells, were observed in the SEM images.

### 2.3. Transmission Electron Microscopy Observations

*E. coli* cells were examined by TEM to characterize the morphological alternations resulting from colistin-plus-tetrandrine treatment. Figure 4A–C reveal a stable structure with a smooth appearance, well-defined intact membranes, and relatively uniform electron density in the cytosol. On the other hand, a destabilized cell wall with a rough surface was observed when bacterial cells were treated with a combination of colistin and tetrandrine (Figure 4D). Furthermore, substantial alterations in shape and integrity or increased permeability, resulting in leaking of cytoplasmic material, were detected. The periplasmic space was distended and occupied by electron-dense material from the cytosol. Several vacuole-like structures were noted where electron density was abridged. There was also a considerable electron-dense region near the transparent regions. Furthermore, some additional membranous structures in the region of cells were found, and the detachment of the inner membrane from the outer membrane was also observed. These morphological alterations reveal the synergistic antibacterial activity of a combination of tetrandrine and colistin, which inhibits the growth of *mcr-1*-positive isolates.

### 2.4. In Vivo Synergistic Activity

To confirm the in vitro results of the above experiments, we evaluated the synergistic potential in vivo in an animal infection model. Mid-log bacteria cultures were suspended to 10^7^ CFU/mL, and 100 μL of suspension was inoculated into the right thigh muscle of neutropenic female mice. One hour later, a placebo (normal saline control), tetrandrine (18.75 mg/kg), and/or colistin (12.5 mg/kg) were administered. In the thighs of the control group, the bacterial loads were 6.5 log10 cfu/g after infection. The bacterial loads in the colistin- and tetrandrine-treated mice groups were 5.6 log10 cfu/g and 5.4 log10 cfu/g, respectively. In contrast, the bacterial load in the colistin-plus-tetrandrine combined-treated group reduced dramatically, which was noted as 2.76 log10 cfu/g (Figure 5). This indicates the synergistic antibacterial activity of a combination of colistin and tetrandrine on *mcr-1*-harboring *E. coli* (*p <* 0.05).

### 2.5. Molecular Docking

The molecular docking results show that tetrandrine can bond with Leu419, Ala420, and Tyr476 by hydrogen bonding, and that the free energy of binding was 8.34 kcal/mol, which was strong evidence of direct physical interaction between tetrandrine and MCR-1 protein (Figure 6).

## 3. Discussion

Antimicrobial resistance is a worldwide health concern, due to which the mortality rate, hospital stay, and treatment costs increase [28]. Before 2015, the most hazardous bacterial-resistance mechanisms were carbapenemase and ESBL production. In the advent of drug resistance, colistin was considered the last treatment option for a cure; colistin has severe side effects in the form of nephro- and neurotoxicity [29,30,31]. The colistin option was in peril when its improper usage was high in human and livestock infection therapy; was also used as a food supplement for poultry and animal farming [32,33,34,35]. Due to its extensive use, there was a continuous worry regarding acquiring the plasmid-mediated resistance against this last drug of choice [36,37,38]. In 2015, the first plasmid-mediated colistin-resistant gene (*mcr-1*) was detected in pig and human origin samples [11]. The detection of this gene represents a step in the post-antibiotic, era because the researcher found that it has horizontal transferability and could transfer to both the same and different species via conjugation. The *mcr-1* gene encodes phosphoethanolamine (pEtN) transferase enzymes that modify the lipid-A portion of the bacterial outer membrane and reduce the affinity of colistin to perform the antibacterial activity [39]. After the development of this resistance, researchers from all over the world started searching for alternative or novel therapeutic options such as bacteriophage therapy, synthetics drugs formulation, and medicinal plant extracts to cure the infections caused by superbugs [40]. 

In the alternative therapeutic options, the extract of medicinal plants is cheap, easy, and has fewer side effects [41]. Tetrandrine is one of the herbal extracts obtained from the roots of *S. tetrandra* plant. Tetrandrine is a plant-based efflux pump inhibitor used to cure cardiovascular disorders as a Ca^+2^ antagonist [42]. Based on its efflux inhibitor potential, tetrandrine is also considered a drug adjuvant against multiple drug-resistant pathogens to improve the retention of drugs inside bacterial cells to perform antibacterial activity. Previous studies have shown that the synergistic activity of tetrandrine with ethidium bromide, fluconazole, and colistin against methicillin-resistant *S. aureus*, *Candida albicans*, and *Salmonella* species [26,43,44]. In the current study, aiming for similar synergism, we analyzed the synergistic activity of tetrandrine with colistin against *mcr-1*-harboring *E. coli*. Based on the literature search and to the best of our knowledge, this is the first study to report the therapeutic activity of tetrandrine against *E. coli*. We performed checkerboard and time-kill assays to determine their antibacterial activity alone and in combination. Our results show that neither colistin nor tetrandrine have antibacterial against *mcr-1* positive *E. coli*. However, when used in combination, the MIC values decreased dramatically. 

Furthermore, FICI indicated that potential synergism existed against all our *mcr-1*-harboring pathogens. Similar trends were also observed from the time-kill assay. We also determined their efficacy in vivo using a neutropenic mouse model. Interestingly, we found that, for the mouse group for which the tetrandrine and colistin were used in combination, the bacterial load was reduced two-fold compared to treatment with either agent alone. A similar in vivo trend has also been observed against *Salmonella*, but, in that case, the bacterial load was measured in the spleen and liver tissue, whereas we measured bacterial content in the thighs of mice [26]. These results indicate that a combination of tetrandrine and colistin could potentially treat localized and systematic infections.

SEM and TEM visualization were carried to observe the effect of tetrandrine and colistin on the morphology of bacterial cells. SEM micrographs of treated cells illustrated remarkable damage to the cell wall, displaying irregularly wrinkled membrane stacks and deep craters. Similarly, TEM images confirmed the disruption of the polar region and leakage of the cytoplasmic content. It has already been reported that the cell-wall degradation is usually caused by the weakening of the peptidoglycan layer [45]. Moreover, we performed a computational analysis of the MCR-1 protein and tetrandrine to find possible direct interactions between the two molecules. It was determined that tetrandrine interacts with three essential amino acids, including Leu419, Ala420, and Tyr476, of MCR-1. Due to this interaction, the MCR-1 might lose its function and affinity toward its substrates, because all three amino acids are located in the active site [46]. These results are consistent with an early report of a combination of tetrandrine and colistin against *Salmonella* species [26]. It is further suggested that tetrandrine and colistin might have the same synergistic effect against all *mcr-1*-harboring Gram-negative pathogens and need to be further investigated. Further studies are required to determine the synergistic activity of colistin and tetrandrine on chromosomal-based resistance to colistin in *E. coli.*

The limitation of the current study is that our *E. coli* were clinically isolates, and no *mcr-1* positive reference study was used in the current study. Moreover, further research work is required to determine the cytotoxicity and bioavailability of a combined tetrandrine and colistin therapy.

## 4. Materials and Methods

### 4.1. Strains, Drugs, and Media

In total, 20 isolates were randomly selected from 75 *mcr-1*-positive isolates detected in our previous study from the fecal samples of healthy livestock in Pakistan [47]. Tetrandrine and colistin sulfate were purchased from the Xi’an Herb Bio-Tech Co., Ltd. (Xi’an, China) and Sigma-Aldrich Trading Co., Ltd. (Shanghai, China). All bacterial media used in this study were purchased from Hope Biotechnology Co., LTD (Qingdao, China).

### 4.2. Antimicrobial Susceptibility Testing and Checkerboard Assay

The minimum inhibitory concentrations (MICs) of colistin and tetrandrine were determined using the broth microdilution method, according to the European Committee on Antimicrobial Susceptibility Testing (EUCAST) guidelines [48]. The checkerboard assay was performed for determining the synergistic activity and fractional inhibitory concentration (FIC) indices of both compounds. Briefly, the 0.5 McFarland turbidity of overnight cultured and tested *E. coli* was achieved and further diluted at 1:100 in a Mueller-Hinton broth (MHB) medium. Subsequently, 100 µL of MHB was added to each well of a 96-well plate. The standard concentrations of tetrandrine (1024 µg/mL) were diluted along the ordinate, while the colistin sulfate was diluted along the abscissa. The plates were incubated at 37 °C for 18–24 h. The MIC of both compounds alone and in combination was determined, and the fractional inhibitory concentration index (FICI) was calculated using the below equation.


(1)
$$\mathrm{FICI}=\frac{\mathrm{MIC}\;\mathrm{in}\;\mathrm{combination}}{\mathrm{MIC}\;\mathrm{alone}}$$


‘Synergy’ was considered to be when FICI ≤ 0.5, 0.5 < FICI ≤ 1 indicates ‘partial synergy’, FICI = 1 indicates an ‘additive effect’, and a FIC index > 1 indicates an ‘antagonistic’ effect [26,49].

### 4.3. Time-Kill Assay

For the time-kill assay, an *mcr-1*-positive strain was randomly selected, and the assay was performed according to the CLSI guidelines. Briefly, the selected strain was cultured overnight, then diluted 1:10,000 in MHB medium and incubated for 4 h at 37 °C to obtain early exponential-phase bacteria. The *E. coli* cells were then treated with tetrandrine (MIC value) and colistin (MIC value), both alone and in combination. The cultures were incubated at 37 °C, and 100 µL of aliquot was removed at 0, 2, 4, 8, and 12 h time points. Then 20 µL of each dilution was inoculated on MHA and incubated at 37 °C for 18–24 h. The number of colonies on each plate was counted. Synergistic activity was interpreted as a 2-log10 decrease in the number of CFU/mL between the combination and its most active single component after 12 h of incubation. 

### 4.4. Fluorescence Staining

For fluorescence staining microscopy, an overnight culture of *mcr-1*-harboring *E. coli* was diluted in 1: 100 in MHB and incubated at 37 °C for 4 h to attain an early exponential growth phase. Then, the culture was divided into three tubes: the colistin (MIC value) tube, the tetrandrine (MIC value) tube, and the combined tube of colistin and tetrandrine. After 18 h of incubation, a small amount of bacterial culture was fixed and stained with fluorescent dyes SYTO 9 and propidium iodide (PI) on the coverslip. Then, the number of live cells was examined under a fluorescence microscope. 

### 4.5. Scanning Electron Microscopy

For SEM analysis, exponentially grown *mcr-1*-positive *E. coli* cells were treated with colistin (MIC value) and tetrandrine (MIC value) alone and in combination for 4 h at 37 °C in Luria broth (LB). The cells were centrifuged and washed twice with phosphate-buffered saline (PBS). The pelleted cells were fixed with 2.5% (*v*/*v*) glutaraldehyde in PBS, incubated overnight at 4 °C, and then serially dehydrated using 30–100% ethanol. After substituting the ethanol with tertiary-butanol, the samples were released onto silver paper for vacuum freeze-drying. Finally, the dried cells were gold-coated and observed under the scanning electron microscope (SEM, Zeiss EVO18, Jena, Germany) at 10.0 kV. 

### 4.6. Transmission Electron Microscopy

For the TEM observations, the *E. coli* cells were prepared as described earlier. The prepared bacterial pellets were subjected to a series of treatments according to the guidelines in the literature in order to perform TEM analysis. Ultrathin sections were prepared on formvar-coated grids (Plano, Wetzlar, Germany) and stained with 3% uranyl acetate. Microscopy was performed with a Hitachi H-9500) (Oberkochen, Germany) microscope at 120-keV electron energy. Zero-loss energy filtering (30) was applied for optimizing the contrast.

### 4.7. In Vivo Synergistic Activity

A female mouse model for *E. coli* infection was used to determine the synergistic effect of tetrandrine in combination with colistin. A total of 20 germ-free mice weighing 25 ± 2 g were treated with 150 mg/kg cyclophosphamide (Shanghai Biolang Biotechnology Co., Ltd, Shanghai, China) for 4 days, followed by a 100 mg/kg dose on the fifth day to induce neutropenia before bacterial inoculation. The mice were divided into four groups, with 5 in each group. The *mcr-1*-harboring *E. coli* isolates were selected for in vivo assay. Mid-log bacteria cultures were diluted to 10^7^ CFU/mL, and 100 μL of suspension was inoculated into the right thigh muscle of each female mouse. After one hour, a placebo (normal saline, Group A), tetrandrine (18.75 mg/kg), and colistin (12.5 mg/kg) were administered in the following manner: tetrandrine only (Group B), colistin only (Group C), and tetrandrine + colistin (Group D). The mice were monitor for 48 h after treatment. 

### 4.8. Molecular Docking of Tetrandrine and MCR-1 Protein

Molecular docking between the crystal structure of the MCR-1 protein (PDB ID: 5GRR) and tetrandrine (PubChem CID: 73078) was performed by AutoDock 4.3 and the Lamarckian Genetic Algorithm method on the Cygwin platform with default parameters to identify any possible direct interactions. The weighting parameters for the recording function included the hydrogen bond energy, hydrophobic interaction, spatial interaction, and the number of rotary keys in the legend. Convergence was measured to evaluate pairing. The lower the parameter, the more likely the ligand will bind to the active site. The docking result was visualized using the PyMOL software for the analysis of binding site residues.

### 4.9. Statistical Analysis

All numerical results are reported as the mean ± SD. The significance of differences between all groups was determined by the Student’s *t*-test and GraphPad Prism v8.0.2. The differences were considered significant at *p* ˂ 0.05.

## 5. Conclusions

In the present study, we determined that tetrandrine has the potential to synergistically enhance the activity of colistin against *mcr-1*-harboring *E. coli* both in vivo and in vitro. Computational analysis showed a direct interaction of tetrandrine with MCR-1 protein at the MCR-1 active site, indicating the ability of tetrandrine to inhibit substrate-MCR-1 binding. This study concluded that tetrandrine is a promising small molecule that can halt *E. coli* superbugs that show resistance to colistin. Further studies are required to determine their activity against chromosomal-based resistance in other colistin-resistant bacterial pathogens. 

## Figures and Tables

**Figure 1 antibiotics-11-01346-f001:**
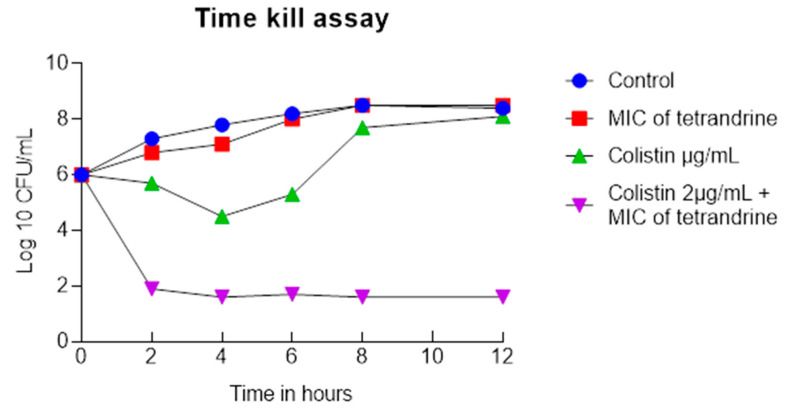
Colistin and tetrandrine in combination inhibits the growth of colistin-resistant *E. coli*. Time kill assay of colistin and tetrandrine alone and in combination shows that neither colistin nor tetrandrine inhibits the growth of *E. coli*. In contrast, the growth of *E. coli* was significantly reduced when treated with a combination of colistin and tetrandrine.

**Figure 2 antibiotics-11-01346-f002:**
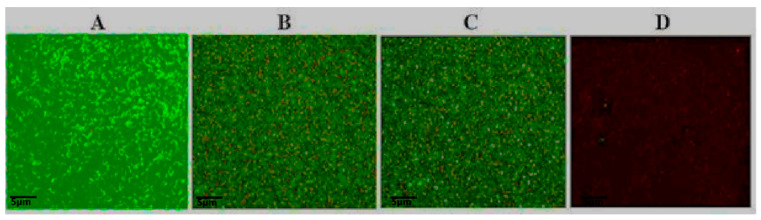
Colistin and tetrandrine, in combination, kill colistin-resistant *E. coli*. Fluorescence microscopic observation of (**A**) control: cells growing in the absence of colistin and tetrandrine. (**B**) Colistin alone does not inhibit the growth of *E. coli*, (**C**) tetrandrine alone does not inhibit the growth of *E. coli*, and (**D**) cells treated with colistin plus tetrandrine in combination show numerous dead cells.

**Figure 3 antibiotics-11-01346-f003:**
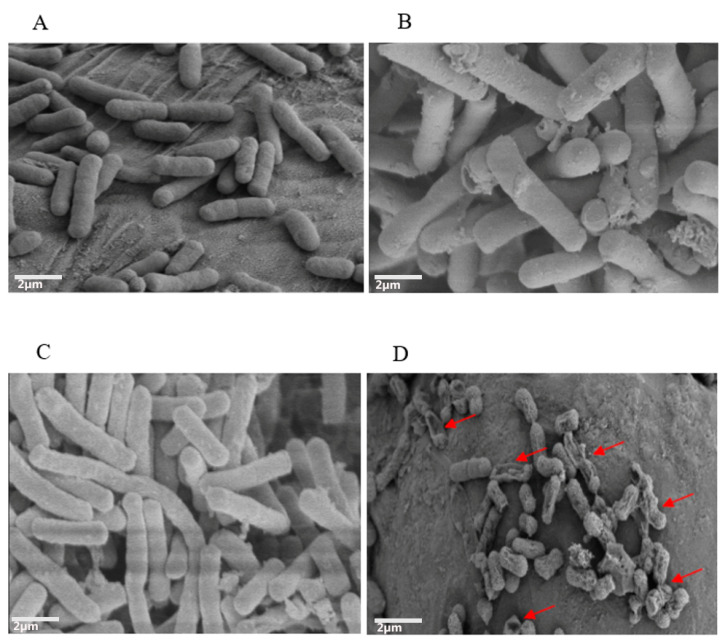
Colistin and tetrandrine in combination causes membrane damage and kills colistin-resistant *E. coli*. Scanning electron microscopic observation of (**A**) control untreated cells showing normal *E. coli* morphology, (**B**) cells treated with colistin alone, (**C**) cells treated with tetrandrine, and (**D**) cells treated with a combination of colistin and tetrandrine showing rupture of outer membrane and leakage of cytoplasmic contents.

**Figure 4 antibiotics-11-01346-f004:**
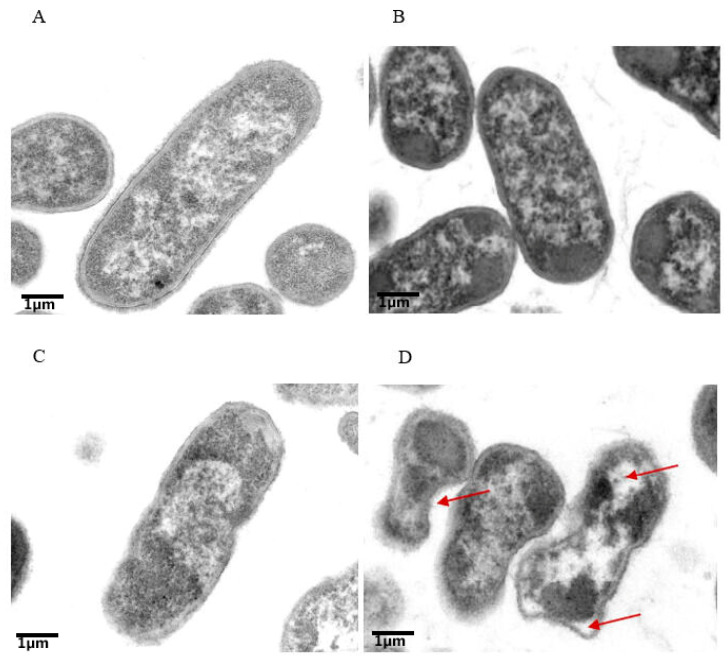
Colistin and tetrandrine in combination causes membrane damage and kills colistin-resistant *E. coli*. Transmission electron microscopic observation of (**A**) control untreated cells with intake cell morphology, (**B**) cells treated with colistin alone, (**C**) cells treated with tetrandrine, and (**D**) Cells treated with a combination of colistin and tetrandrine showing discontinuation of membranes and vacuoles moving out of cells.

**Figure 5 antibiotics-11-01346-f005:**
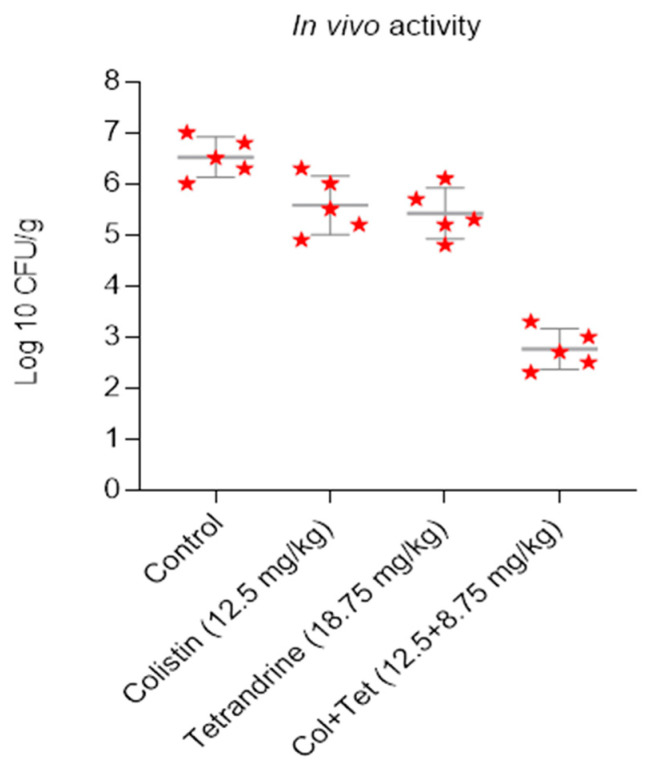
Colistin and tetrandrine together inhibit infection by colistin-resistant *E. coli* in neutropenic mice. In vivo activity of colistin and tetrandrine alone and in combination in neutropenic mice, infected with colistin-resistant *E. coli*, show that neither colistin nor tetrandrine alone inhibits the growth of *E. coli*, but treatment with colistin and tetrandrine in combination reduces the bacterial load.

**Figure 6 antibiotics-11-01346-f006:**
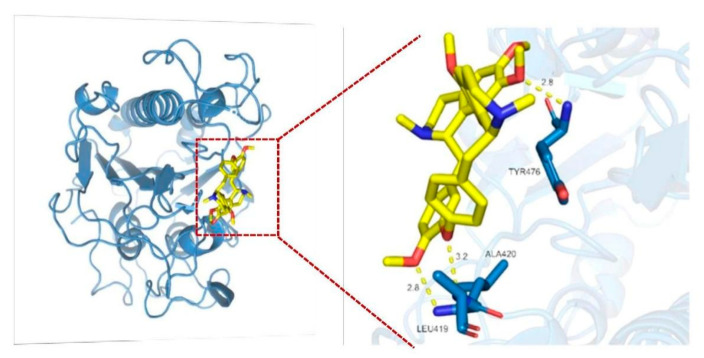
Molecular docking of tetrandrine and MCR-1 protein show that tetrandrine forms hydrogen bonding with three amino acids of MCR-1 protein in its active sites, suggestive of inhibition of MCR-1 in terms of colistin resistance.

**Table 1 antibiotics-11-01346-t001:** Susceptibility values of colistin and tetrandrine alone and in combination. The table represents the MIC, FICI, and fold change of colistin MIC values in a total of 20 tested *E. coli* strains.

Strain Number	COL MIC (µg/mL)	TET MIC (µg/mL)	TET+COL MIC (µg/mL)	FICI	COL MIC (Fold Change)
1	2	320	1	0.5	≥32
2	2	640	1	0.187	≥32
3	4	640	1	0.187	≥128
4	4	640	1	0.156	≥128
5	4	640	1	0.187	≥128
6	4	640	1	0.141	≥128
7	4	1280	2	0.141	≥64
8	8	1280	2	0.125	≥128
9	16	640	4	0.156	≥256
10	4	320	2	0.141	≥64
11	4	320	2	0.266	≥64
12	4	320	2	0.141	≥4
13	4	320	2	0.156	≥64
14	4	1280	1	0.156	≥64
15	2	320	0.5	0.156	≥32
16	4	320	2	0.141	≥128
17	4	1280	2	0.25	≥128
18	4	640	2	0.25	≥128
19	4	640	1	0.187	≥128
20	4	1280	1	0.125	≥64

COL; colistin, TET; tetrandrine, MIC; minimum inhibitory concentration, FICI; fractional inhibitory concentration index.

## Data Availability

Not applicable.

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
