# Peer review of "Synergistic Activity of Tetrandrine and Colistin against *mcr-1*-Harboring *Escherichia coli"

_antibiotics, 2022, doi:10.3390/antibiotics11101346_

Round 1
Reviewer 1 Report (Previous Reviewer 1)
On account of the manuscript ANTIBIOTICS-1954118, entitled “Synergistic activity of tetrandrine and colistin against mcr-1-harboring Escherichia coli” by Muhammad Shaf et al., this manuscript was newly submitted to the same Journal after revisions of the previous manuscript of ANTIBIOTICS-1897075, entitled “Synergistic activity of tetrandrine and colistin against mcr-1 harboring Escherichia Coli”. The topic is important to better understanding of the mcr-1-harboring E. coli, and to conduct environmental health risk management of livestock in Pakistan as well. The authors revised the manuscript appropriately according to the Reviewers comments. After careful consideration, I made a decision that the manuscript is acceptable for publication in its present form.
Author Response
We appreciate the Editors' and reviewers' warm work earnestly. Thank you for reviewing our manuscript and for accepting it to be publish in the Journal of Antibiotics. We are hopeful that the results of this manuscript will be interesting to the readers.
Reviewer 2 Report (Previous Reviewer 4)
The authors have addressed comments and incorporated the recommended changes.
Author Response
We appreciate the Editors' and reviewers' warm work earnestly. Thank you for reviewing our manuscript and for accepting it to be publish in the Journal of Antibiotics. The final version of manuscript was edited by two professional English editors. We are hopeful that the results of this manuscript will be interesting to the readers.
This manuscript is a resubmission of an earlier submission. The following is a list of the peer review reports and author responses from that submission.
Round 1
Reviewer 1 Report
On account of the manuscript ANTIBIOTICS-1897075, entitled “Synergistic activity of tetrandrine and colistin against mcr-1 harboring Escherichia Coli” by Muhammad Shafiq et al., the authors evaluated the antibacterial synergistic activity of tetrandrine in combination with colistin against mcr-1-harboring E. coli. In addition, the authors are investigated the computational analysis of MCR-1 protein and tetrandrine to determine interaction interface of these two molecules. The topic is important to better understanding of the mcr-1-harboring E. coli, and to conduct environmental health risk management of livestock in Pakistan as well. After careful consideration, I feel that this manuscript is to be published after improvement of some minor shortcomings. Details of my comments are as follows:
1) The view point of this research is interesting, and the authors got interesting results. Several revisions are, however, required before publication. The first one is in that the present Abstract was not informative. Abstract should include purpose of the research, principal results and major conclusions in a summarized way with using actual values, etc. In addition, due to separation of the Abstract from the major article, it must be a key to lead readers to evoke a spirit of challenge to contact with the contents of the report, as described in Author instructions of this Journal. Therefore, the authors are encouraged to improve the Abstract.
2) Another notable aspect is in the Introduction. The aim and novelty of the present study is not so clear. This causes the difficulty of the readers to follow up precisely the results and discussion that the authors aimed at the present report. The authors don't necessarily mention general issues in detail, but are better to show information in a summarized way with focusing on the main issues related to the originality, necessity, and usefulness of the contents of the present investigation clearly, which surpass the previous reviews. Otherwise, novelty of this work on tetrandrine and colistin against mcr-1 harboring Escherichia Coli would not be strengthened. The authors are strongly encouraged to take these aspects into account to show the results for enhancement of the novelty and better understanding of the results. After that I am ready to recommend the present manuscript for publication.
Author Response
We thank you and found your comments and critique very helpful that can be strengthened to increase the readability and interpretabilitiy of results for readers. We have addressed these comments to the best of our efforts, and all changes in the manuscript have been highlighted in red color in the revised manuscript. Our point to point response to comments are outlined below.
Reviewer 1:
On account of the manuscript ANTIBIOTICS-1897075, entitled “Synergistic activity of tetrandrine and colistin against mcr-1 harboring Escherichia Coli” by Muhammad Shafiq et al., the authors evaluated the antibacterial synergistic activity of tetrandrine in combination with colistin against mcr-1-harboring E. coli. In addition, the authors are investigated the computational analysis of MCR-1 protein and tetrandrine to determine interaction interface of these two molecules. The topic is important to better understanding of the mcr-1-harboring E. coli, and to conduct environmental health risk management of livestock in Pakistan as well. After careful consideration, I feel that this manuscript is to be published after improvement of some minor shortcomings. Details of my comments are as follows:
Comment 1: The view point of this research is interesting, and the authors got interesting results. Several revisions are, however, required before publication. The first one is in that the present Abstract was not informative. Abstract should include purpose of the research, principal results and major conclusions in a summarized way with using actual values, etc. In addition, due to separation of the Abstract from the major article, it must be a key to lead readers to evoke a spirit of challenge to contact with the contents of the report, as described in Author instructions of this Journal. Therefore, the authors are encouraged to improve the Abstract.
Response 1: Thank you for reviewing our work. The abstract section is modified accordingly and the been highlighted in red color.
Comment 2: Another notable aspect is in the Introduction. The aim and novelty of the present study is not so clear. This causes the difficulty of the readers to follow up precisely the results and discussion that the authors aimed at the present report. The authors don't necessarily mention general issues in detail, but are better to show information in a summarized way with focusing on the main issues related to the originality, necessity, and usefulness of the contents of the present investigation clearly, which surpass the previous reviews. Otherwise, novelty of this work on tetrandrine and colistin against mcr-1 harboring Escherichia Coli would not be strengthened. The authors are strongly encouraged to take these aspects into account to show the results for enhancement of the novelty and better understanding of the results. After that I am ready to recommend the present manuscript for publication.
Response 2: Thank you for pointing out this; we have revised the Introduction section carefully according to your suggestions. We again appreciate your suggestions. We have addressed these comments to the best of our efforts, and all changes were made accordingly. We hope that you will find the revised manuscript suitable for publication in the Journal of Antibiotics.
Reviewer 2 Report
This study provided evidence of combination of colistin and tetrandrine to combat mcr-harboring E. coli. However, some revision required to improve the manuscript below
1. Introduction: Usually, colistin will use for highly MDR such as carbapenem-resistant organisms. It will be better if the author could extend the paragraph on lines 48 to highlight the important of colistin such as "Nowaday carbapenem-resistant Enterobacterales harboring mcr have been reported worldwide (ref*), therefore, colistin is extensively used to treatment those of infections that lead to increasing of the colistin resistance"
*reference to be cited
1.1. Huang H, Dong N, Shu L, et al. Colistin-resistance gene mcr in clinical carbapenem-resistant Enterobacteriaceae strains in China, 2014-2019. Emerg Microbes Infect. 2020;9(1):237-245.
1.2. Kim JS, Yu JK, Jeon SJ, et al. Distribution of mcr genes among carbapenem-resistant Enterobacterales clinical isolates: high prevalence of mcr-positive Enterobacter cloacae complex in Seoul, Republic of Korea. Int J Antimicrob Agents. 2021;58(5):106418.
1.3. Paveenkittiporn W, Kamjumphol W, Ungcharoen R, Kerdsin A. Whole-Genome Sequencing of Clinically Isolated Carbapenem-Resistant Enterobacterales Harboring mcr Genes in Thailand, 2016-2019. Front Microbiol. 2021;11:586368.
1.4. Bastidas-Caldes C, de Waard JH, Salgado MS, et al. Worldwide Prevalence of mcr-mediated Colistin-Resistance Escherichia coli in Isolates of Clinical Samples, Healthy Humans, and Livestock-A Systematic Review and Meta-Analysis. Pathogens. 2022;11(6):659.
2. Line 76-77: I do not understand what is the author meaning? Increase or decrease?
3. Table 1: What is MIC of combination? I recommend the author please see "Lopez-Carrizales M, Velasco KI, Castillo C, et al. In Vitro Synergism of Silver Nanoparticles with Antibiotics as an Alternative Treatment in Multiresistant Uropathogens. Antibiotics (Basel). 2018;7(2):50." to correct the author's Table 1 as Table 2 of the suggested paper.
In addition, the author should show Figure of checkerboard assay similar in Figure 4 of the suggested paper. A schematic picture is OK.
4. Figure 1: What is MIC value of tetradrine?
5. Line 205: Upper case of 's' for staphylococcus.
6. M & M: subtopic 4.1: What is the criteria used to random 20 mcr-isolates from 75 isolates?
7. M & M: subtopic 4.2, the author should extend description of checkerboard assay in more detail, please see an example of the suggested paper above. In addition, reference need to be cited for checkerboard assay.
And version of CLSI should be specified, is it M100, M07 or another?
8. M & M: subtopic 4.7, how long the author monitor mouse after treatment? Additional detail should be explained.
9. M & M: Subtopic 4.9, the author mentioned about statistical analysis, but no description in the result, where? What is experiment shown statistic significant?
Author Response
Reviewer 2:
This study provided evidence of combination of colistin and tetrandrine to combat mcr-harboring E. coli. However, some revision required to improve the manuscript below
Comment 1. Introduction: Usually, colistin will use for highly MDR such as carbapenem-resistant organisms. It will be better if the author could extend the paragraph on lines 48 to highlight the important of colistin such as "Nowaday carbapenem-resistant Enterobacterales harboring mcr have been reported worldwide (ref*), therefore, colistin is extensively used to treatment those of infections that lead to increasing of the colistin resistance"
*reference to be cited
1.1. Huang H, Dong N, Shu L, et al. Colistin-resistance gene mcr in clinical carbapenem-resistant Enterobacteriaceae strains in China, 2014-2019. Emerg Microbes Infect. 2020;9(1):237-245.
1.2. Kim JS, Yu JK, Jeon SJ, et al. Distribution of mcr genes among carbapenem-resistant Enterobacterales clinical isolates: high prevalence of mcr-positive Enterobacter cloacae complex in Seoul, Republic of Korea. Int J Antimicrob Agents. 2021;58(5):106418.
1.3. Paveenkittiporn W, Kamjumphol W, Ungcharoen R, Kerdsin A. Whole-Genome Sequencing of Clinically Isolated Carbapenem-Resistant Enterobacterales Harboring mcr Genes in Thailand, 2016-2019. Front Microbiol. 2021;11:586368.
1.4. Bastidas-Caldes C, de Waard JH, Salgado MS, et al. Worldwide Prevalence of mcr-mediated Colistin-Resistance Escherichia coli in Isolates of Clinical Samples, Healthy Humans, and Livestock-A Systematic Review and Meta-Analysis. Pathogens. 2022;11(6):659.
Response 1: Thank you for reviewing our manuscript. The needful suggestion has been done and the suggested references have been added in revised manuscript. Line # 48-50.
Comment 2. Line 76-77: I do not understand what is the author meaning? Increase or decrease?
Response 2. Line # 66, the word maximizes is replaced by “increase”
Comment 3. Table 1: What is MIC of combination? I recommend the author please see "Lopez-Carrizales M, Velasco KI, Castillo C, et al. In Vitro Synergism of Silver Nanoparticles with Antibiotics as an Alternative Treatment in Multiresistant Uropathogens. Antibiotics (Basel). 2018;7(2):50." to correct the author's Table 1 as Table 2 of the suggested paper. In addition, the author should show Figure of checkerboard assay similar in Figure 4 of the suggested paper. A schematic picture is OK
Response 3: Table 1 is modified accordingly. However, we don’t have the picture for checkerboard.
Comment 4. Figure 1: What is MIC value of tetradrine?
Response: The MIC value of tetrandrine is 320–640 µg/mL
Comment 5. Line 205: Upper case of 's' for staphylococcus.
Response 5: Modified accordingly.
Comment 6. M & M: subtopic 4.1: What is the criteria used to random 20 mcr-isolates from 75 isolates?
Response 6: 20 colistin-resistant mcr-1 harboring isolates were randomly selected because all were mcr-1 positive and colistin-resistant. Not some specific criteria were applied. It was a long run to process all 75 isolates, therefore only 20 of them were processed for this experiment as the resistant profiles of all were the same.
Comment 7. M & M: subtopic 4.2, the author should extend description of checkerboard assay in more detail, please see an example of the suggested paper above. In addition, reference need to be cited for checkerboard assay. And version of CLSI should be specified, is it M100, M07 or another?
Response 7: Thanks for your suggestion. The citation has been added to the manuscript. The minimum inhibitory concentrations (MICs) of colistin and tetrandrine were determined by the broth microdilution method, according to the European Committee on Antimicrobial Susceptibility Testing (EUCAST) guidelines
Comment 8. M & M: subtopic 4.7, how long the author monitor mouse after treatment? Additional detail should be explained.
Response 8: The mouse was monitored for 48 h after treatment. Line# 319
Comment 9. M & M: Subtopic 4.9, the author mentioned about statistical analysis, but no description in the result, where? What is experiment shown statistic significant?
Response 9: The combined synergistic activity was statistically significant (P< 0.05). added to line # 158.
Reviewer 3 Report
The topic of the article is of novelty and of high clinical significance, given the current increasing antibiotic resistance in bacteria worldwide. The manuscript is well written and designed. The methods are new and appropriate for this study. However, I would have few suggestions, to imrove the quality of the article:
- in the title, please change the C from Escherichia Coli to small case letter, as it is written in the rest of the manuscript
- please, add a section about the study limitations, in which to include the fact that there was no mcr-1 positive reference strain used and to specify that further studies are required in order to test for citotoxicity and bioavailability.
Author Response
Reviewer 3:
Comment 1: The topic of the article is of novelty and of high clinical significance, given the current increasing antibiotic resistance in bacteria worldwide. The manuscript is well written and designed. The methods are new and appropriate for this study. However, I would have few suggestions, to imrove the quality of the article:
- in the title, please change the C from Escherichia Coli to small case letter, as it is written in the rest of the manuscript
Response 1: Thank you for highlighting it. The needful suggestion has been made in the revised manuscript.
Comment 2:- please, add a section about the study limitations, in which to include the fact that there was no mcr-1 positive reference strain used and to specify that further studies are required in order to test for citotoxicity and bioavailability.
Response 2: The study limitations have been added to the revised manuscript as suggested line # 251-253.
Reviewer 4 Report
The study describes the synergistic effect of tetrandrine and colistin against mcr-1 harboring E. coli. The study's findings are interesting, but the manuscript should make some recommendations.
1. The manuscript needs editing for English grammar and formatting.
2. Line 1: Delete the line "Type the Paper" and also remove the brackets for "Article."
3. Line 2: The word "Coli" should be in small letters "coli"
4. Line 22: The phrase "Our results showed that neither colistin" is a scientifically well-established finding. How this manuscript can say, their results show this. This should be deleted from the manuscript.
5. Line 29-30: Abstract conclusion should focus on the current study's findings instead of future directions.
6. Line 38-39: The phrase "production due to various genes like CTX-M, TEM, and SHV" is unclear. If the authors describe the genes, they should be "production due to various genes like blaCTX-M, blaTEM, and blaSHV" or they should mention them as proteins.
7. Line 40-41: Several mechanisms are involved in carbapenem resistance. I recommend rewriting the sentence because just mentioning "but due to their improper use" is not enough to support the resistance. I suggest looking into and citing the resistance mechanism from a local study https://doi.org/10.1038/s41598-019-38943-7
8. Line 60: The phrase "Based on its promising synergistic activity" is not scientific as the synergistic efficacy of tetrandrine and colistin has not yet well-established. It is suggested to delete this.
9. Line 64-65: The phrase "The finding of this study will provide an alternative therapeutic option for colistin-resistant E. coli infections" does not make sense as the study's findings are not yet established. In conclusion, the authors emphasized conducting further studies to determine their activity against the other colistin-resistant bacterial pathogens. The lines need rephrasing.
10. The study's aims are unclear and need to be revised for greater clarity.
11. Line 243: Despite having 75 isolates, only 20 isolates were included in the study. How did the authors select the sample size, and what criteria did they follow to select the strains? It should be clearly mentioned in the manuscript.
12. Line 256-257: The phrase "The standard concentrations of tetrandrine were diluted" needs clarity and should be elaborated.
13. The authors have used chunks of methods from the following article without citing them. This should be cited in the methodology https://doi.org/10.1016/j.biopha.2022.112873.
14. The manufacturers' names, cities, and countries should be mentioned for the chemical, kits, and instruments. A few of them have missing information like manufacturer or city etc.
15. Line 321: Which version of Graph Pad Prism was used? The software company name, city, and country should be included. The figures show that the authors have used several other software not mentioned in this section.
16. The study has several limitations, which should be described in the discussion section.
17. The citations do not well support the study. The number of references is insufficient to support the introduction, methodology, and discussion, which needs consideration. The authors should include the following recent studies in the introduction to see the burden of AMR on humans and animals.
https://doi.org/10.3390/antibiotics10101263; https://doi.org/10.3390/antibiotics10040467; https://doi.org/10.1371/journal.pone.0245126
18. The authors have not written sufficient discussion concerning the local Pakistani studies, and also, what about the use of other combinations against the mcr-1? Following local and other studies should be included in the discussion section.
https://doi.org/10.3389/fmicb.2019.02957; https://doi.org/10.1186/s13099-020-00392-3; https://doi.org/10.3390/ijerph191610449; https://doi.org/10.1016/j.ijid.2021.03.004
Author Response
Reviewer 4:
The study describes the synergistic effect of tetrandrine and colistin against mcr-1 harboring E. coli. The study's findings are interesting, but the manuscript should make some recommendations.
Comment 1: The manuscript needs editing for English grammar and formatting.
Response 1: As per suggestion, we have edited our manuscript by two professional editors. All the revised changes have been highlighted in red color in the revised manuscript.
Comment 2: Line 1: Delete the line "Type the Paper" and also remove the brackets for "Article."
Response 2: The needful suggestion has been made
Comment 3: Line 2: The word "Coli" should be in small letters "coli"
Response 3: Modified accordingly
Comment 4: Line 22: The phrase "Our results showed that neither colistin" is a scientifically well-established finding. How this manuscript can say, their results show this. This should be deleted from the manuscript.
Response 4: Thanks for your suggestion. We have modified the sentence in the revised manuscript.
Comment 5: Line 29-30: Abstract conclusion should focus on the current study's findings instead of future directions.
Response 5: The future direction statement is omitted.
Comment 6: Line 38-39: The phrase "production due to various genes like CTX-M, TEM, and SHV" is unclear. If the authors describe the genes, they should be "production due to various genes like blaCTX-M, blaTEM, and blaSHV" or they should mention them as proteins.
Response 6: The word “bla” has been added
Comment 7: Line 40-41: Several mechanisms are involved in carbapenem resistance. I recommend rewriting the sentence because just mentioning "but due to their improper use" is not enough to support the resistance. I suggest looking into and citing the resistance mechanism from a local study https://doi.org/10.1038/s41598-019-38943-7
Response 7: Sentence has been modified and the citation has been added in the revised manuscript.
Comment 8: Line 60: The phrase "Based on its promising synergistic activity" is not scientific as the synergistic efficacy of tetrandrine and colistin has not yet well-established. It is suggested to delete this.
Response 8: removed accordingly
Comment 9: Line 64-65: The phrase "The finding of this study will provide an alternative therapeutic option for colistin-resistant E. coli infections" does not make sense as the study's findings are not yet established. In conclusion, the authors emphasized conducting further studies to determine their activity against the other colistin-resistant bacterial pathogens. The lines need rephrasing.
Response 9: The suggestion has been done accordingly “Findings of this study could provide a baseline for further research work in order to establish an alternative therapeutic option for treating colistin-resistant E. coli infections.” Line # 70-72.
Comment 10: The study's aims are unclear and need to be revised for greater clarity.
Response 10: Revised the aims now as mentioned in the response of comment 9
Comment 11: Line 243: Despite having 75 isolates, only 20 isolates were included in the study. How did the authors select the sample size, and what criteria did they follow to select the strains? It should be clearly mentioned in the manuscript.
Response 11: Twenty colistin-resistant mcr-1 harboring isolates were randomly selected because all were mcr-1 positive and colistin-resistant. Not some specific criteria were applied. It was a long run to process all 75 isolates, therefore only 20 of them were processed for this experiment as the resistant profiles of all were the same.
Comment 12: Line 256-257: The phrase "The standard concentrations of tetrandrine were diluted" needs clarity and should be elaborated.
Response 12: The sentence is modified accordingly “The standard concentrations of tetrandrine (1024 µg/mL) were diluted along the ordinate, while the colistin sulfate was diluted along the abscissa”.
Comment 13: The authors have used chunks of methods from the following article without citing them. This should be cited in the methodology https://doi.org/10.1016/j.biopha.2022.112873.
Response 13: Added in method section as suggested
Comment 14: The manufacturers' names, cities, and countries should be mentioned for the chemical, kits, and instruments. A few of them have missing information like manufacturer or city etc.
Response 14: The manufacturers' names, cities, and countries have been mentioned for the chemical, kits, and instruments in the revised manuscript as suggested.
Comment 15: Line 321: Which version of Graph Pad Prism was used? The software company name, city, and country should be included. The figures show that the authors have used several other software not mentioned in this section.
Response 15: added in revised manuscript.
Comment 16: The study has several limitations, which should be described in the discussion section.
Response 16: The limitation section is added at the end of discussion. Line #251-253.
Comment 17: The citations do not well support the study. The number of references is insufficient to support the introduction, methodology, and discussion, which needs consideration. The authors should include the following recent studies in the introduction to see the burden of AMR on humans and animals.
https://doi.org/10.3390/antibiotics10101263; https://doi.org/10.3390/antibiotics10040467; https://doi.org/10.1371/journal.pone.0245126
Response 17: Added in the revised manuscript
Comment 18: The authors have not written sufficient discussion concerning the local Pakistani studies, and also, what about the use of other combinations against the mcr-1? Following local and other studies should be included in the discussion section.
https://doi.org/10.3389/fmicb.2019.02957; https://doi.org/10.1186/s13099-020-00392-3; https://doi.org/10.3390/ijerph191610449; https://doi.org/10.1016/j.ijid.2021.03.004; https://doi.org/10.1186/s12879-021-05906-1
Response 18: All the suggested references have been added in the revised manuscript.